# Prospective cohort study to investigate the burden and transmission of acute gastroenteritis in care homes: a study protocol

Thomas Inns,[1,2,3] Anna Pulawska-Czub,[2,4] John P Harris,[1,2] Roberto Vivancos,[2,3,5] Jonathan M Read,[2,5,6] Nicholas J Beeching,[2,5,7,8] David J Allen,[2,9] Miren Iturriza-Gomara,[2,4] Sarah J O'Brien[1,2,5]

For numbered affiliations see end of article.

**Correspondence to**
Thomas Inns;
thomas.inns@liverpool.ac.uk

## ABSTRACT

**Introduction** Noroviruses are the leading cause of acute gastroenteritis in all age groups, but illness is more severe and causes excess mortality in the elderly, particularly those in long-term care. The total burden of norovirus disease in the elderly in the UK is poorly defined; no current surveillance programmes systematically or accurately quantify norovirus infection in those living in care homes. The aim of this study is to evaluate an enhanced surveillance system for acute gastroenteritis among the elderly in care homes.

**Methods and analysis** We will conduct this prospective cohort study in care homes in North West England; residents and staff at study care homes will be asked to participate. We will prospectively enrol a cohort of participants in an enhanced surveillance system to capture the incidence of acute gastroenteritis and use multiplex PCR to detect pathogens. We will sample symptomatic and non-symptomatic participants to understand characteristics of norovirus disease and susceptibility to infection. We will generate novel data on transmission dynamics by collecting data on the pattern of interactions within care homes using electronic proximity sensors. Comparisons of outbreak and non-outbreak periods will be used to quantify the impact of norovirus outbreaks on care homes.

**Ethics and dissemination** The study has been approved by the North West–Greater Manchester South NHS Research Ethics Committee (REC Reference: 16/NW/0541). Study outputs will be disseminated through scientific conferences and peer-reviewed publications. This study will provide detailed insight on the burden and aetiology of acute gastroenteritis in care homes, in addition to generating novel data on transmission dynamics and risks. The study will identify areas for improving infection control practice and allow more accurate modelling of the introduction of interventions such as vaccination.

## Strengths and limitations of this study

► This prospective cohort study will provide detailed information on the incidence, risk factors for and aetiology of acute gastroenteritis in care homes.

► This study will use novel methods to generate data on the structure of interactions between persons in care homes.

► The results of this study will provide clinicians and policy-makers with information to improve infection control and prevention in care homes.

► The results of this study will inform modelling to estimate the burden of acute gastroenteritis in care homes in the UK and the potential impact of a norovirus vaccine targeted at this population.

► It may be that participants at study sites systematically differ from others at care homes both in the UK and in other countries in aspects such as the demography, socioeconomic status and level of morbidity of participants.

gastroenteritis is generally mild and self-limiting, but there is increasing evidence that it may lead to long-term sequelae[4 5] and contribute to excess mortality in the elderly and the immunocompromised.[6–13] The elderly, particularly those in long-term care, suffer a longer duration of illness with more severe symptoms, contributing to excess mortality.[6 7]

The factors that facilitate sustained transmission in health and social care settings are likely to be the result of a combination of the environment, behaviour patterns associated with patients, visitors and staff, the characteristics of the norovirus strains and/or host-related factors that influence susceptibility to disease.[14 15] At present, the main approaches to preventing and controlling norovirus outbreaks, common across several national guidelines, include promotion of hand hygiene, patient isolation (separation

## INTRODUCTION

Noroviruses are endemic in the human population and are recognised as the leading cause of infectious intestinal disease across all ages.[1–3] In healthy populations, norovirus

of symptomatic patients) and cohorting (grouping of patients based on symptoms), staff exclusion from work, visitor restrictions, enhanced environmental cleaning and disinfection, and closures of units.[16–20]

The total burden of norovirus disease in the elderly population in the UK is poorly defined, despite the widespread acknowledgement that the elderly and, in particular, those in long-term care are worst affected by norovirus illness. There are currently no surveillance programmes that can systematically or accurately quantify the levels of any cause of norovirus-specific gastroenteritis among the growing ageing population living in care homes. The ageing population in the UK means that those over 65 years old are the fastest growing sector of the population, which will result in increasing pressure and demand for healthcare and long-term residential care in the future.[21] Norovirus infections are known to be more severe in this sector of the population, contributing to excess hospitalisation and mortality.[6 22–25]

The relative importance of different drivers of transmission and factors that impact on susceptibility to disease and more severe illness among this population are poorly understood. Norovirus infections of the elderly are associated with prolonged shedding and longer duration of symptoms,[26 27] and it has been proposed that the elderly may contribute to the emergence of new epidemic strains that spread across the population.[28]

## AIMS AND OBJECTIVES

The aim of this study is to evaluate an enhanced surveillance system for acute gastroenteritis among the elderly in care homes. This will provide data that can then be extrapolated and used in mathematical models to calculate the burden of norovirus infections in the elderly in long-term residential care in the UK and the potential impact of a norovirus vaccine specifically targeted to this population.

### Study objectives

1. To study the feasibility of using an enhanced acute gastroenteritis surveillance system in care homes to generate novel descriptive data regarding norovirus infection in this population;
2. To quantitatively assess the impact of norovirus illness on residential care institutions;
3. To generate novel data on transmission dynamics and risks, by collecting both data on the pattern of interactions within care homes and data on virus characterisation;
4. To understand characteristics of norovirus disease and susceptibility to infection (viral load, shedding duration, norovirus-specific IgA antibodies, blood group and microbiota composition (Diversity Index)) and use this to inform transmission dynamics studies, by sampling symptomatic and non-symptomatic participants;

5. To understand the risk factors associated with acquiring norovirus infection in residential care settings during a norovirus outbreak.

## METHODS AND ANALYSIS

### Study setting and location

The study will take place in care homes in North West England. Care homes are defined as places offering accommodation and personal care for people who may not be able to live independently. This includes nursing homes that offer the same type of care but with the addition of 24-hour medical care from a qualified nurse.

The North West of England has a population of over 7 million people. Within the region, there is a mixture of affluent and deprived areas, urban and rural. The study sites will be in the metropolitan boroughs of Liverpool and Sefton. The combined population of these two boroughs is 746 604, of which 130 458 (17.47%) are aged 65 or older.[29] Within the two boroughs, there are 133 care homes registered with the Care Quality Commission.[30]

### Sampling frame and strategy

The sampling frame is the total number of residential care homes for the elderly in the metropolitan boroughs of Liverpool and Sefton, registered with the Care Quality Commission. The sampling strategy is a convenience sample of sites that are approached and agree to participate. We will aim to recruit four study sites prospectively.

### Study overview and study design

The study will be based on a prospective cohort of participants in the enhanced surveillance system (component A). The other six study components are outlined in table 1 and include different epidemiological, microbiological and quantitative elements.

### Study site inclusion process

Potential study sites will be recruited in two ways. Study sites will be recruited prospectively to be included in all study components. These sites will be approached through the Liverpool Community Health Trust (LCHT). We will

| **Table 1** Summary of study components | |
| --- | --- |
| **Study component** | **Study component description** |
| A | Enhanced surveillance system |
| B | Pathogen testing |
| C | Individual norovirus risk factor study |
| D | Microbiota as a risk factor for norovirus infection and/or disease |
| E | Transmission dynamics study |
| F | Norovirus outbreak risk factors |
| G | Quantitative assessment of the impact of norovirus outbreaks |

approach care homes after discussion with LCHT so that those approached are representative of the care homes in the sampling frame (eg, in size and complexity). Study sites will also be recruited reactively. Potential study sites within the sampling frame will report to the Public Health England (PHE) Health Protection Team that they are experiencing an outbreak of gastroenteritis (as required by The Health and Social Care Act 2008). If this reported outbreak meets the study definition, PHE will inform the study team who will contact the study site to ask them to participate. Two members of the study team are substantive employees of PHE; their job roles entail surveillance of infectious disease outbreaks, and they will, therefore, be informed of relevant outbreaks. Potential study sites that are reactively recruited will be asked to participate in the study components B, D, F and G. Studies recruited both prospectively and reactively will have background information collected on the type of residents, structure, capacity and staffing at the care home.

### Study sample size

Data from PHE surveillance of gastrointestinal illness in care homes in Cheshire and Merseyside indicate that the median number of residents is 32 and the median number of staff is 35. On the basis of these estimates, it would be expected that there are an average of 67 participants per study site. The study will aim to recruit four study sites prospectively. It is therefore expected that approximately 268 participants will be included in all components of the study.

### Enhanced surveillance system (component A)

The study population will be residents and staff at study sites who have provided informed consent. Cases are defined as persons in the study population with the following:

► Vomiting—two or more episodes of vomiting in a 24-hour period OR
► Diarrhoea—three or more loose stools in a 24-hour period OR
► Vomiting AND diarrhoea—one or more episodes of BOTH symptoms in a 24-hour period

Confirmed cases will be defined as:

► Cases with a positive laboratory diagnosis of an infectious cause

Causes of diarrhoea and vomiting should be believed to be infectious. Non-infectious causes, which are not to be counted, would include long-standing diarrhoea associated with disability or incontinence, and ingestion of laxative drugs.

Current numbers of residents and staffing levels at each care home will be collected using a questionnaire, filled in by a member of staff in conjunction with a research nurse on the first Monday of each month. Faecal samples will be obtained for each case to determine whether the illness is caused by norovirus or another gastrointestinal pathogen; this sample will be collected as soon as possible after onset of illness. For each case, information

including onset date, medical history, duration of symptoms, complications, hospitalisation and outcome (eg, death) will be collected using a questionnaire. This questionnaire will be filled in by a member of staff on the same day as the faecal sample is collected, then checked and collected by a research nurse on the first Monday of each month.

We will describe the characteristics of the surveillance system and the epidemiology of cases it captures. Non-cases will contribute to the person-time at risk. We will calculate incidence rates for person-time at risk in each study site and for the study in total. We will compare the burden of norovirus, viral gastroenteritis of another cause and gastroenteritis of an unknown cause. We will describe the duration and severity of illness.

### Pathogen testing (component B)

Stool samples will be collected for cases as described for the enhanced surveillance system. The specimen request form will be completed and the sample and request form will be submitted to a laboratory to be tested for gastrointestinal pathogens. Samples will be posted using approved containers for the transport of diagnostic specimens. Samples will be sent to Liverpool Clinical Laboratories, based in the Royal Liverpool University Hospital.

Diagnostic tests will be done in real time and results reported to the study team. Samples will be tested for 15 pathogens using Luminex xTAG Gastrointestinal Pathogen Panel. Positive results will be reported to the study team and copied to the patient's general practitioner. The operation of this study will not interfere with public health actions. A portion of each virus positive sample will be sent for genotyping and further characterisation. This work will not be conducted in real time and will not lead to public health action.

We will describe the proportion positive for norovirus and for other infectious causes of acute gastroenteritis. We will describe the sequencing results of norovirus-positive samples over time, by study site and in relation to the results of the enhanced surveillance system.

### Individual norovirus risk factor study (component C)

For each case with a norovirus positive laboratory result, where feasible, sequential stool samples will be taken at three time points (from onset): days 0–3, days 6–8 and days 12–15. For each sample, viral load will be measured by qRT-PCR.

For the purpose of testing for blood group, all participants enrolled prospectively in the enhanced surveillance system will be asked to provide a sample of saliva. Samples will be taken by research nurses within a month of consent being obtained. Laboratory analysis will use one enzyme immunoassay (EIA) for the detection of blood group antigens in saliva and another EIA for the detection of norovirus-specific IgA and IgG in saliva. Results will be standardised measuring total IgA.

We will compare proportion and level of cases with viral shedding over time. We will categorise study participants

by blood group category and compare case incidence rates (based on person-time at risk) by pathogen.

## Microbiota as a risk factor for norovirus infection and/or disease (component D)

All participants enrolled prospectively in the enhanced surveillance system will be asked to provide a stool sample at the beginning of the study. Samples will be taken by research nurses within a month of consent being obtained. If study participants test positive for norovirus and provide a sequential stool sample, as detailed in study component C, this sample will also be used for microbiota investigations, in addition to the baseline sample.

Stool samples arriving for microbiota analysis will have norovirus viral load measured by qRT-PCR. Samples for microbiota analysis will be stored frozen for a maximum of 2 weeks prior to DNA extraction. Samples will be treated by bead beating and lysozyme prior to DNA extraction using a commercially available extraction kit. We will store two aliquots of each stool samples at −80°C, adding nucleic acid stabilising buffer to one of them. Stool DNA extraction will be conducted from the aliquots in stabilising buffer, in batches, once or twice a month.

DNA samples will undergo metataxonomic analysis with 16S rDNA in the first instance. An aliquot of DNA will be stored for further metagenomic analysis, subject to obtaining additional funding. The 16S rDNA PCR strategy will use a nested dual index protocol to amplify and barcode the variable V3–V4 region (319f—5′ ACTCCTACGGGAGGCAGCAG 3′ and 806r—5′ GGACTACHVGGGTWTCTAAT 3′) resulting in ~469 bp PCR product.[31] Then barcoded 16S PCR products will be multiplexed and run in batches of up to 96 on the MiSeq to produce 2×300 bp reads.

Sequences generated on MiSeq will undergo a validated error correction protocol. We will trim the start and end of the reads based on quality scores using a tool such as Sickle (https://github.com/najoshi/sickle), error correction with BayesHammer[32] followed by overlapping reads with PANDAseq[33] with a minimum overlap of 10 bp for V3/V4 reads. These corrected and overlapped reads will then be analysed with QIIME.[34] USEARCH will be run using de novo and open reference operational taxonomic unit (OTU) clustering methods, and de novo chimara detection conducted with software such as UCHIME. Taxonomy will be assigned to OTUs using the naive Bayesian RDP Classifier using both the SILVA and GREENGENES taxonomic databases.

Estimates of within-sample species richness (number of OTUs) and diversity (Shannon Index) at multiple rarefaction depths will be compared between cases and controls, and between samples obtained from cases during the acute and convalescent phases, using Student's t-test.

Weighted and unweighted Unifrac will be used to measure distances in microbiota composition between these groups. These results will be visualised using principal coordinates analyses and statistically significant clusters identified using *adonis*. Random Forest regression will be used to identify OTUs that distinguish norovirus cases and diarrhoeal or asymptomatic controls and between samples obtained from cases during the acute and convalescent phases.

## Transmission dynamics study (component E)

We will quantify potential transmission paths into and within care homes using a survey instrument (an individually worn electronic proximity sensor) that has previously proved successful in characterising interaction patterns in a study of influenza in an American school and in other settings.[35–37]

The study population will be participants at care homes enrolled in the enhanced surveillance system and visitors to those homes on the days of data collection. Visitors on the days in question will be consented using a specific consent form. Four 24-hour study periods will be chosen for the transmission dynamics study. The 24-hour periods selected will be a convenience sample based on study team availability and study site access. Interactions between individuals will be quantified using electronic proximity sensors, called motes, which are worn by or located next to participants and detect and record the nearby presence of other motes. The age, gender and role of individual in the care home will be collected. A member of the study team will be present at the study site of the days of data collection to consent visitors and distribute the motes.

We will measure continuous, uninterrupted interactions between participants. We will sum the total durations of interactions over each 24-hour period and create participant contact networks with edge weights proportional to the total number of mutual interactions. We will use social network analysis and transmission models to relate infection attack rates and dynamics to measured interaction patterns.

## Norovirus outbreak risk factors (component F)

One of the objectives of this study is to understand the risk factors associated with acquiring norovirus infection in care homes during a norovirus outbreak. An outbreak will be defined as two or more cases (as defined in component A) occurring in an institution, with onset of illness within 5 days. An outbreak will be considered finished if no new cases are ascertained for 7 days. An outbreak-free period will be defined as a period of time ending 3 weeks before an outbreak is declared and beginning 3 weeks after an outbreak is considered over in an institution.

Participants enrolled at a study site where an outbreak has occurred will form a cohort in which we will investigate risk factors associated with infection. A member of the study team will administer a questionnaire to participants; the questionnaire covers demographic, illness and medication information, along with food and drink history and information on time spent in different areas of the study site.

We will compare patient characteristics and exposure histories between cases and non-cases, with the null

hypothesis that risk factors have a similar distribution in cases and non-cases. We will investigate differences using univariable and multivariable analyses.

## Quantitative assessment of the impact of norovirus outbreaks (component G)

A case-crossover approach will be used to compare resource usage and operational efficiency in residential institutions during outbreak periods and outbreak-free periods (as defined in component F). Data collected will include the symptoms of ill staff, the days of work lost, the need for additional staff (bank or agency) and additional cleaning. Operational impact data collected will include isolation of residents, transfers to healthcare facilities and blocking/delays on places to new residents. These data will be collected by a member of the study team from a member of the affected care home.

Data will be collected for the whole period when an outbreak is occurring. The increased resource usage during outbreaks will be measured by comparing periods when outbreaks are occurring with periods when there is not an outbreak occurring. Measurement in outbreak-free periods will take place 3 weeks after the end of outbreaks so that activity will resettle to normal levels. We will test the null hypothesis that norovirus outbreaks do not have an impact on the resource usage of care homes.

## ETHICS AND DISSEMINATION
### Consent
Informed consent will be obtained for each participant with capacity to consent. For those persons without capacity to consent, a nominated person who meets the criteria described in Section 32 of the Mental Capacity Act 2005 will be asked to provide consent.

### Timeline
Administrative and logistical arrangements have been made to start the study in February 2017 and collect data for a 2-year period.

### Dissemination of findings
Study results will be presented and discussed at appropriate scientific meetings, and published in open access peer-reviewed journals. Appropriate metadata will be published with the research data to enable other researchers to identify whether the data could be suitable for their own research.

**Author affiliations**
[1]Institute of Psychology Health and Society, University of Liverpool, Liverpool, UK
[2]NIHR Health Protection Research Unit in Gastrointestinal Infections, University of Liverpool, Liverpool, UK
[3]Field Epidemiology Services, Health Protection, Public Health England, Liverpool, UK
[4]Institute of Infection and Global Health, University of Liverpool, Liverpool, UK
[5]NIHR Health Protection Research Unit in Emerging Infections and Zoonoses, University of Liverpool, Liverpool, UK
[6]Centre for Health Informatics Computing and Statistics, Lancaster Medical School, Lancaster University, Lancaster, UK
[7]Tropical and Infectious Disease Unit, Royal Liverpool and Broadgreen University Hospitals NHS Trust, Liverpool, UK
[8]Clinical Sciences Group, Liverpool School of Tropical Medicine, Liverpool, UK
[9]Department of Pathogen Molecular Biology, Faculty of Infectious and Tropical Diseases, London School of Hygiene and Tropical Medicine, London, UK

**Contributors** TI participated in the design of the study, will oversee the study coordination, data collection and analysis, and wrote the manuscript. AP-C will contribute to study coordination, data collection and analysis. JPH participated in the design of the study and will contribute to study coordination and data collection and analysis. RV conceived the study, participated in the design of the study and will contribute to study coordination, data collection and analysis. JMR participated in the design of the study and will contribute to data collection and analysis. NJB participated in the design of the study and will facilitate data collection. DJA participated in the design of the study and will contribute to data analysis. MI-G conceived the study, participated in the design of the study and will contribute to study coordination, data collection and analysis. SJOB conceived the study, participated in the design of the study and will contribute to study coordination, data collection and analysis.

**Funding** The research is funded by the National Institute for Health Research Health Protection Research Unit (NIHR HPRU) in Gastrointestinal Infections at University of Liverpool in partnership with Public Health England (PHE), in collaboration with University of East Anglia, University of Oxford and the Institute of Food Research. Thomas Inns is based at the University of Liverpool. The views expressed are those of the author(s) and not necessarily those of the NHS, the NIHR, the Department of Health or Public Health England.

**Competing interests** None declared.

**Ethics approval** The study has been approved by the North West–Greater Manchester South NHS Research Ethics Committee (REC Reference: 16/NW/0541). The study is sponsored by the University of Liverpool.

**Provenance and peer review** Not commissioned; externally peer reviewed.

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
