## [Reviewer comments · BMJ Open]

ARTICLE DETAILS

TITLE (PROVISIONAL)	A prospective cohort study to investigate the burden and transmission of acute gastroenteritis in care homes: a study protocol
AUTHORS	Inns, Thomas; Pulawska-Czub, Anna; Harris, John; Vivancos, Roberto; Read, Jonathan; Beeching, Nicholas; Allen, David; Iturriza-Gomara, Miren; O'Brien, Sarah

VERSION 1 – REVIEW

REVIEWER	Pat Stone Columbia University, USA
REVIEW RETURNED	24-Aug-2017

GENERAL COMMENTS	In abstract state study design (prospective observational?). Currently it says “we will conduct the study...”. Strengths and limitations do not discuss the novel methods or state study design. It is not clear what inclusion or exclusion criteria will be (has been) used to recruit the for “study sites”. How many individuals do the investigators think will be included—that is what are the expected sample sizes for all components of the study? Recruiting sites “reactively” doesn’t sound prospective. There are inconsistencies. The sentences under “study overview and study design” section are not clear? It is confusing to discuss component A, before all the components have been listed. Can these components be mapped to the 5 study aims? At least provide a brief overview in the intro-aims section Component A-while cases are well articulated. There is no discussion of controls (non-cases). This is needed. Component E, transmission dynamics study-it is not clear how visitors will be consented into the study and wear the “motes”. Can this be clarified. Minor, use comma before “which”
---

REVIEWER	Mary Wi Centers for Disease Control and Prevention, United States ksw
REVIEW RETURNED	28-Aug-2017

GENERAL COMMENTS	This will be a very interesting study, and the results should go far in informing future norovirus work.
--

VERSION 1 – AUTHOR RESPONSE

Reviewer: 1

Reviewer Name: Pat Stone

Institution and Country: Columbia University, USA Competing Interests: None declared

Comment: In abstract state study design (prospective observational?). Currently it says “we will conduct the study...”.

Response: We thank the reviewer for this comment and have added this to the methods and analysis section of the abstract.

Comment: Strengths and limitations do not discuss the novel methods or state study design.

Response: We have added the suggested information to the strengths and limitations bullet points.

Comment: It is not clear what inclusion or exclusion criteria will be (has been) used to recruit the for “study sites”. How many individuals do the investigators think will be included—that is what are the expected sample sizes for all components of the study?

Response: We thank the reviewer for this comment. In the “Sampling frame and strategy” section of the methods and analysis (page 5, lines 16-20) we outline that sites will be recruited as a convenience sample. Regarding the inclusion criteria, we have a section on the “Study site inclusion process” (page 5, lines 28-43). We have added a section on the “Study sample size”, as in the REC-approved protocol, to address the reviewer’s comments on this topic.

Comment: Recruiting sites “reactively” doesn’t sound prospective. There are inconsistencies.

Response: In the manuscript we state that this is predominantly a prospective cohort study, however due to the unpredictable nature of acute gastroenteritis outbreaks we have the ability to recruit sites reactively for some elements of the study.

Comment: The sentences under “study overview and study design” section are not clear? It is confusing to discuss component A, before all the components have been listed. Can these components be mapped to the 5 study aims? At least provide a brief overview in the intro-aims section Component A-while cases are well articulated. There is no discussion of controls (non-cases). This is needed.

Response: We note the reviewer’s comments and have added a small table to the “study overview and study design” section, to list all the components as suggested. We have added a sentence to the section on Component A and text to Component C to clarify that non-cases will contribute to person-time at risk in the enhanced surveillance component and individual norovirus risk factor component.

Comment: Component E, transmission dynamics study-it is not clear how visitors will be consented into the study and wear the “motes”. Can this be clarified.

Response: We have added text to the “Transmission dynamics study (Component E)” section to clarify this point.

Comment: Minor, use comma before “which”

Response: We have reviewed the use of “which” in the manuscript, and have either deleted or changed to “that”. Where it is retained it is used in line with standard British English.

Reviewer: 2

Reviewer Name: Mary Wikswo

Institution and Country: Centers for Disease Control and Prevention, United States

Competing Interests: None declared

Comment: This will be a very interesting study, and the results should go far in informing future norovirus work.

Response: We thank the reviewer for this comment.